# The Evolution of the Role of External Ventricular Drainage in Traumatic Brain Injury

**DOI:** 10.3390/jcm8091422

**Published:** 2019-09-10

**Authors:** Charlene Y. C. Chau, Claudia L. Craven, Andres M. Rubiano, Hadie Adams, Selma Tülü, Marek Czosnyka, Franco Servadei, Ari Ercole, Peter J. Hutchinson, Angelos G. Kolias

**Affiliations:** 1Division of Neurosurgery, Department of Clinical Neurosciences, Addenbrooke’s Hospital and University of Cambridge, Cambridge Biomedical Campus, Cambridge CB20QQ, UK; 2Victor Horsley Department of Neurosurgery, National Hospital for Neurology and Neurosurgery, Queen Square, London WC1N3BG, UK; 3Neurosciences Institute, INUB-MEDITECH Research Group, El Bosque University, 113033 Bogotá, Colombia; 4NIHR Global Health Research Group on Neurotrauma, University of Cambridge, Cambridge CB20QQ, UK; 5Department of Neurosurgery, Innsbruck Medical University, 6020 Innsbruck, Austria; 6Department of Neurosurgery, Humanitas University and Research Hospital, 20090 Milan, Italy; 7Division of Anaesthesia, Addenbrooke’s Hospital and University of Cambridge, Cambridge Biomedical Campus, Cambridge CB20QQ, UK

**Keywords:** neurosurgery, ventriculostomy, neurotrauma, intracranial pressure, EVD, TBI, ICP

## Abstract

External ventricular drains (EVDs) are commonly used in neurosurgery in different conditions but frequently in the management of traumatic brain injury (TBI) to monitor and/or control intracranial pressure (ICP) by diverting cerebrospinal fluid (CSF). Their clinical effectiveness, when used as a therapeutic ICP-lowering procedure in contemporary practice, remains unclear. No consensus has been reached regarding the drainage strategy and optimal timing of insertion. We review the literature on EVDs in the setting of TBI, discussing its clinical indications, surgical technique, complications, clinical outcomes, and economic considerations.

## 1. Introduction

Traumatic brain injury (TBI) remains a global public health challenge [1]. It is a major cause of death and disability across all ages worldwide, with significant socio-economic repercussions. The rising global incidence of TBI is attributable to the increase in road traffic collisions and trauma-related violence in low- and middle-income countries (LMICs) [2]. In high-income countries (HICs), the introduction of traffic safety regulations and education, as well as an ageing population, have shifted the demographics of head trauma [2]. A national UK trauma database identified the changing trend in the typical TBI patient profile, from a young male individual, to that of an older adult, more commonly female, as a result of falls [3]. 

Following a primary injury, brain oedema, disturbed cerebrospinal fluid (CSF) dynamics, and the presence of mass lesions exhaust the volume buffering compensatory mechanism mediated by CSF drainage from brain cisterns to the spinal subarachnoid space, and can lead to a rise in intracranial pressure (ICP) [4,5,6,7]. The subsequent increase in ICP and impaired autoregulation result in reduced cerebral perfusion, exacerbating cerebral ischaemia and secondary brain injury [8]. Brain tissue compression, distortion, and herniation can subsequently occur, ultimately leading to death [9,10]. Therefore, control of ICP is an important factor in the medical and surgical management of TBI. 

The potential benefits of EVD placement following TBI are three-fold: (1) Drainage of even small volumes can lower ICP significantly; (2) clearance of haemorrhage from a ventricle, thus preventing subsequent hydrocephalus; and (3) enabling monitoring of ICP via the pressure transducer vent port, providing objective information to guide ICP/cerebral perfusion pressure (CPP)-directed therapies [11].

Despite these potential benefits, the effectiveness of CSF drainage in improving outcomes for TBI patients is unclear, given the varying level of patency of the ventricles and ICP response to drainage [12]. Despite new recommendations on EVDs in recent TBI guidelines, the low level of evidence has, in part, led to heterogeneity in usage, drainage strategy, and timing [13]. 

We review herein the topic of EVDs in the setting of TBI, summarising available evidence regarding the following: (1) Historical evolution as a procedure; (2) indications for EVD insertion: ICP monitoring; (3) indications for EVD insertion: CSF diversion; (4) surgical technique; (5) complications; (6) clinical outcomes; (7) economic considerations; and (8) directions for future research. 

## 2. Methods

This was a narrative review, thus a PRISMA (Preferred Reporting Items for Systematic Reviews and Meta-Analyses) flowchart was not necessary. We searched MEDLINE (Ovid), EMBASE, The Cochrane Central Register of Controlled Trials (CENTRAL, The Cochrane Library, Latest Issue), Scopus, and Web of Science for English language full-text articles published from 1995 to 2019, using three search terms combined with the Boolean operator “AND”. The search terms were “traumatic brain injury”, “external ventricular drains”, and “raised ICP”. Relevant articles were read, and their references were screened for further relevant articles. Selection of articles for inclusion in our review was based on a combination of research quality, high number of citations, and current and potential impact on clinical practice.

## 3. Literature Review 

### 3.1. Historical Overview 

Knowledge of CSF dates back to the Old Kingdom of Egypt (3000–2500 BC). The Edwin Smith Surgical Papyrus first documented the fluid in the brain [14]. Cerebrospinal fluid was perceived as pathological “moistness” by Hippocrates (460–375 BC), and took on a theological sense as Galen (130–200) described it as an excretionary liquid for “pneumas” (spirit) or a “spirituous lymph” by Swedenborg (1688–1772) [14,15]. The early 18th century saw a paradigm shift in how the production and circulation of CSF was viewed, with the term “liquid cephalo-spinal” first introduced by Magendie in 1825 [14]. 

The procedure of external ventricular drainage was first reported in the 18th century to treat congenital hydrocephalus (Figure 1). Claude-Nicholas Le Cat (1700–1768) documented his procedure of ventricular puncture with an adapted catheter and trocar from those used to treat ascites [16]. However, early attempts yielded no success [16,17]. Ventricular drainage was later deemed by Robert Whytt (1714–1766) to expedite death, complementing the remarks of Benjamin Hill that it should not be used even as a last resort [17].

Improvements in the ventricular drainage technique were observed in the mid 19th century, establishing the viability of ventricular drainage. This was attained by three separate advancements: (i) The incorporation of the aseptic technique, (ii) an awareness of the consequences of excessive ventricular drainage, and (iii) identification of optimal sites for catheter insertion [17]. These paved the way for improvements in apparatus and system setup during the 20th century. In 1941, the first closed sterile system for ventricular drainage was described by Ingraham and Campbell [18]. The attention to sterility is evident from the technical description of autoclaving or sterilising the instruments, and “fluffs” around tubing [18]. Their stopcock also addressed the problem of over-drainage, allowing a slow and controlled CSF withdrawal [17,18]. Later innovations also sought to improve the safety of the procedure: They included a drip chamber to block the backflow of contaminated CSF, and flexible silicone catheters that allowed subcutaneous tunnelling [17,19]. The advent of the volumetric EVD via the LiquoGuard^®^ device can also address over-drainage effectively [20].

The enhanced safety and usability diversified indications for EVDs. The added benefit conferred by EVDs, by coupling ventricular pressure monitoring and drainage, which was pioneered by Franklin Robinson in 1948, further encouraged EVD use in patients with non-traumatic neurological conditions, such as subarachnoid haemorrhage, brain tumours, and Reye’s syndrome characterised by cerebral oedema [17,21,22]. The use of EVDs for ICP monitoring in patients with severe head injury, and the relationship between ICP and neurological outcomes, were also increasingly reported in case series [23,24,25,26]. This led to the wide adoption of EVDs in TBI management protocols, as an ICP monitor and CSF drainage apparatus [22]. Nevertheless, concerns that EVD-related complications cause significant patient morbidity and mortality persisted throughout the years, along with controversies regarding the drainage method (continuous or intermittent) [27,28].

### 3.2. Indications for External Ventricular Drains in TBI: Intracranial Pressure Monitoring

#### 3.2.1. Current International Practice

External ventricular drains can be used not only as a therapeutic tool to control raised ICP but also as a monitoring tool. Practice variation regarding the use of EVDs as a monitoring tool exists. In a survey of 66 European neurotrauma centres, the majority indicate that EVDs are utilised in cases where CSF drainage is anticipated or where ventricles are enlarged, and routinely use IPMs [29]. As for the USA, Colombia, and China, external ventricular drains are the preferred monitoring modality [29,30,31,32].

#### 3.2.2. Gold Standard for ICP Monitoring

External ventricular drains have often been considered the gold standard for ICP monitoring [33]. This may in part be a reflection of historical convention since it was the first and only monitor available for many years since the description first given by Guillaume and Janny in 1951, and later Lundberg in 1960 [21,34,35]. Lundberg connected the ventricular catheter to a strain-gauge pressure transducer, with signals amplified electronically and recorded on a paper recorder [21]. This is analogous to modern practice, whereby one of the extracranial ends is attached to an external strain gauge pressure sensing membrane (draining end should be closed for a proper measurement). Conventionally, the ICP measurements via ventriculostomy are deemed more accurate. Reasons are two-fold: (i) It reflects average ICP over the (anterior) intracranial system, as the CSF pressure should equilibrate with the ventricular CSF spaces and is not affected by shearing forces; and (ii) it allows in situ zeroing through resetting the transducer to atmospheric pressure at the level of the tragus or the Foramen of Monro [33,36]. This pressure measurement is sometimes referred to as pCSF.

#### 3.2.3. ICP Thresholds in TBI

The relationship between the extent of intracranial hypertension and mortality is well documented [37,38,39,40]. The ICP threshold beyond which treatment for intracranial hypertension is usually initiated in patients with severe TBI ranges from 15 to 25 mmHg in the existing literature [39,41,42,43]. A systematic review investigating the association between ICP values and patterns and outcomes demonstrated that raised ICP between the ranges of 20 and 40 mmHg is associated with death by an odds ratio (OR) of 3.5, whilst ICP > 40 mmHg had an OR of 6.9 [44]. Apart from the absolute values of ICP, neurological outcomes may be better predicted by the response to management and secondary indices, such as pressure–volume compensatory reserve and cerebrovascular reactivity [45]. ICP refractory to medical management dramatically increases the relative risk of death (OR 114.3) [44].

#### 3.2.4. Problems with ICP Monitoring from EVDs

Intracranial pressure gradients between intracranial compartments may exist in TBI pathologies, for instance, intracranial masses or obstructed CSF flow, challenging the former assumption that the ICP obtained is of “global” representation [46]. The capability to recalibrate may be subjected to human errors when levelling the transducer to the zero-reference point [46]. Altered head positioning when patients are moved may also be overlooked, with the transducer position not corrected [47]. Whilst reliable ICP values depend on timely recalibration, such judicious adjustments may not always be attainable. Rather, recalibration is guided by the hospital protocol, which is not standardised globally or nationally; it is often done after patient transport, during care shift, or when confronted by questionable ICP values or waveforms [48]. 

There is a risk of collapsed ventricles or mechanical obstruction producing potentially erroneous results. Furthermore, a recent systematic review investigating the effectiveness of EVDs versus IPMs for ICP monitor-guided treatment found no difference in mortality or functional outcomes, but higher overall complication rates in those who received an EVD [49]. The weak body of evidence, mainly observational studies with serious or critical risk of bias, precludes any definitive conclusion to be drawn.

Although beyond the scope of this review, there is of course the controversy over the utility of ICP monitoring in TBI. Cohort studies have delineated the association of ICP monitoring and better outcome [7,50,51]. However, retrospective reports have shown worsened survival, prolonged ventilation, and increased therapy intensity in patients with their ICPs being monitored [52,53]. A randomised controlled trial (BEST TRIP trial) comparing an ICP-based monitoring strategy with one reliant on imaging and clinical examination revealed no significant differences in primary outcome [54]. However, given the bias towards neuropsychological performance in the primary composite endpoint of 21 components, the 5% non-significant decrease in mortality and increase in favourable outcomes may have been overlooked [55]. There were also some benefits, including more efficient care, to those who underwent ICP monitoring after TBI [56]. Of note, the study centres in Bolivia and Ecuador used IPM reportedly due to the instrument’s safety profile, ease of insertion, and low maintenance [54]. Furthermore, many devices can also measure pulsatility and compliance, important parameters that are often overlooked in TBI studies but are often important in conditions of abnormal CSF dynamics. Additionally, continuous recording of the ICP and arterial blood pressure waveforms can be used to provide information on the state of cerebrovascular reactivity (PRx index) and can be used to estimate optimal cerebral perfusion pressure levels for individual patients [57,58].

In the past, draining CSF and measuring ICP simultaneously produced artefacts, with the ICP value dependent more on the outflow pressure than the actual pressure within the cranium (Figure 2) [59]. The three-way stopcock would be turned “off” to the drain, whilst “on” for the pressure transducer. Some patients required insertion of an ICP probe, for continuous reliable un-interrupted ICP monitoring [60]. 

However, over the past decade, the advent of more advanced EVD devices and drainage systems have enabled simultaneous drainage and monitoring. Such devices include double lumen ventricular cathethers—Liquoguard machines^®^ (Möller Medical GmbH & Co, Fulda, Germany) and Raumedic EVDs (Raumedic AG, Helmbrechts, Germany) [61,62,63]. Retrospective analysis comparing it against a conventional EVD demonstrated concordance in ICP values and waveforms [64]. 

#### 3.2.5. Future Research Directions 

The latest guidelines from the Brain Trauma Foundation (BTF) no longer addressed the topic of intracranial pressure technology, as technology assessment was deemed to be beyond its scope [13]. However, technological advancements in both intraparenchymal and EVD-based ICP monitoring can improve the safety and functionality of both devices. Further research on these newer technologies is required. 

Advances in biomedical engineering technologies, such as microRNA and MALDI mass spectroscopy, are rapidly expanding our knowledge of diagnostic and prognostic biomarkers that can be detected in the CSF [65,66]. Apart from proteomic profiling, several known biomarkers found from TBI CSF include injury biomarkers from neuronal cells (e.g., Ubiquitin C-terminal hydrolase-L1), astroglia (e.g., glial fibrillary acidic protein), and oligodendrocytes (e.g., myelin basic protein) [65,67,68]. Such biomarkers detected in the CSF could enable more effective triaging, and could become a neuro-monitoring tool for patients with TBI managed in an ICU setting. However, there is a need for further well-designed multi-centre studies and trials to be conducted before they become part of routine clinical practice [69]. 

### 3.3. Indications for External Ventricular Drains in TBI: Cerebrospinal Fluid Drainage

#### 3.3.1. Theoretical Indication for EVDs in TBI

The therapeutic rationale of CSF drainage is encompassed by the Monro–Kellie doctrine. The doctrine stipulates that three intracranial components—brain parenchyma (85%), cerebral vasculature (10%), and CSF (5%)—exist in a pressure–volume equilibrium, such that an increase in the volume of any one of the components should be compensated by a decrease in the other two [70,71]. However, in patients with TBI with exhausted intracranial compliance, even small increments in volume lead to an exponential increase in ICP. CSF drainage, even a few millilitres, therefore, acts to rapidly bring the patient back to the “flat part” of the sigmoidal ICP–intracranial volume (IV) curve from the decompensated phase (Figure 3). CPP (CPP = MAP − ICP) would also theoretically increase for any constant MAP. In addition, studies have confirmed the continuity between CSF and interstitial fluid (ISF); thus, CSF drainage would allow the removal of excess ISF and inflammatory biomarkers, both of which are associated with oedema and ischaemia [72]. 

#### 3.3.2. Current International Practice

The recent update of the BTF guidelines found that the evidence to support the use of CSF drainage in TBI patients is limited and of low quality [13]. A tiered approach, which is widely adopted for TBI management, refers to a combination of various treatments, which are administered in a stepwise manner [74]. The higher the tier, the more deleterious the side effects associated with the interventions. In a TBI survey of 66 European neurotrauma centres, CSF drainage demonstrated the largest within-region variation [29]. Most centres in the Baltic States and Eastern Europe indicate that CSF drainage is institutionally considered a first-tier treatment for elevated ICP in their hospital policy, with 20% and 17% of participants stating it as a second-tier intervention, respectively. Northern, Western, and Southern Europe preferentially use CSF drainage as a second-tier treatment of elevated ICP, whilst the United Kingdom (UK) and Israel exclusively consider the use of CSF drainage as a second-tier therapy [29]. 

The large heterogeneity reflects the uncertainty over the benefits and harms of CSF drainage, with the judgement reliant on physician preference, resource availability, and healthcare policies [29]. However, a close examination of the existing literature revealed that the tiers are defined differently. Some guidelines consider “tier zero” as the starting point, which includes basic intensive care unit (ICU) measures, such as sedation and analgesia for intubation or head-of-bed elevation [75]. Others regard them as first-line measures [12,75,76]. Apart from inconsistencies in tier definition, the composition and ICP thresholds differ. Escalation to a higher tier is characterised by a failure to control a pre-defined ICP or CPP threshold. As such, even if CSF drainage is categorised in the same tier, the ICP threshold to initiate a tier of therapy differs between centres, with ICP thresholds defined as >15, >20, >25, >30 mmHg, or individualized [29]. 

#### 3.3.3. Effect on Intracranial Pressure

Studies investigating the effect of EVD on intracranial pressure reported a reduction in ICP following drainage, but the extent and the sustainability of the decrease differs [76,77,78,79,80,81,82]. Apart from the heterogeneity in TBI patterns and patient profiles, the drainage details and the individual patient’s response to drainage may serve to justify these disparities. 

First, the ICP thresholds for initiating drainage vary. A study in the United States (US) suggested the threshold of ICP as >15 mmHg for intracranial hypertension, with ventriculostomy as the initial therapeutic intervention [77]. Others suggested ICP ≥ 20 mmHg, with different proposed durations, such as >5 min or >10 min [76,78,79,80]. CSF drainage was also used in patients with ICP >25 mmHg for an hour, as a last-tier therapy alternative to decompressive craniectomy [81]. Different CPP thresholds are also reported [12,82]. 

Second, there is clinical equipoise over continuous and intermittent drainage, the latter either for a certain duration or with the amount determined by the treating physician. The two strategies represent the focus over the monitoring (intermittent) and drainage (continuous) component [79,83]. The continuous drainage strategy has shown to result in a lower overall ICP burden, as well as lower levels of biochemical markers related to secondary injury in certain subpopulations [28,79]. ICP may then be only intermittently monitored unless an additional continuous method of measurement is employed. However, there are concerns over ventricular collapse and EVD obstruction with the continuous strategy, which could be mitigated if CSF drainage is done intermittently. In the field of aneurysmal subarachnoid haemorrhage, a recent study found a significant reduction in ventriculoperitoneal (VP) shunt rates after changing to an intermittent CSF drainage with a rapid EVD wean approach (13% intermittent/rapid vs. 35% continuous/gradual, OR 0.21, *p* = 0.001) [84]. It is possible that intermittent drainage may lead to early recruitment of normal CSF outflow pathways. In theory, a closed EVD and relatively higher CSF compartment pressures could facilitate CSF resorption through arachnoid granulations, thereby reducing the need for a VP shunt [85]. However, this has not been proven and the findings of the study by Rao et al. cannot be necessarily extrapolated to the field of TBI. 

Third, patients’ responses to CSF drainage may be disparate. A study investigating CSF drainage in TBI patients illustrates a differing response in the 24 h after EVD insertion [12]. A total of 46% (n = 24) failed to have a sustained decrease in ICP, with ICP values exceeding 20 mmHg within 24 h. This group of patients also did not show clinically significant improvements in CPP, cranio-spinal compensation, and cerebral oxygenation. The authors reported a trend towards a larger ventricular size in those with persistent ICP control, which may suggest a larger CSF volume is responsible for the intracranial hypertension, but there was no statistically significant difference between the groups. Patient demographics, and physiological and radiological variables were also not significantly different, but the radiological information only classified patients with diffuse injury or mass lesions on computed tomography (CT) scans. In addition, the drainage strategy, and the volume of CSF drained, may influence results.

#### 3.3.4. Effect on Other Physiological Parameters

Improvements in some physiological parameters were also reported following drainage in a few studies. Cerebral perfusion pressure and cerebral oxygenation (PbtO_2_) improved, whilst other perfusion indices, such as cerebral blood flow volume (CBFV) and regional cerebral oxygenation (rSO_2_), did not [12,76,77,78,80]. This may be related to the choice of parameters, particularly the sensitivity, resolution, and correct location for monitoring [78]. Brain tissue oxygenation has demonstrated a small but significant improvement post-drainage in two observational studies [12,80]. Authors hypothesise that these observations could be a result of improved cerebral metabolism, or a decrease in pulse amplitude on ICP waveforms and an increased intracranial compliance brought about by the reduced ICP [12,80]. 

#### 3.3.5. Effect on Functional Outcome 

Only two retrospective cohort studies have examined the relationship between CSF drainage and mortality as its primary outcome [86]. The findings of the study by Griesdale et al. showed that EVD usage was associated with an odds ratio of 2.8 and 2.1 for hospital and 28-day mortality, respectively. The authors partially adjusted for the confounding by indication using logistic regression models, as subjects with EVDs have lower GCS and CT characteristics suggestive of higher ICP. However, residual confounding may remain. The small sample size and the nature of the single-centre study also limit the external validity of this study. In addition, the study only included mortality in the short-term but not other functional outcomes at other time points. The other study by Bales et al. also demonstrated that EVD usage led to higher propensity-score weighted odds (OR 2.46) for in-hospital mortality. In addition, these patients had a lower extended Glasgow Outcome Scale at 6 months. However, the primary goal of the study was to compare EVD and IPM as ICP-monitoring devices, not the therapeutic impact of CSF drainage. As such, the authors used a restricted window of EVD placement within 6 h of hospital arrival for patient selection.

#### 3.3.6. Future Research Directions 

The “when” of CSF drainage in TBI patients has not received much attention. A systematic review (unpublished data provided by the authors) investigating the optimal timing of external ventricular drainage identified a lack of studies with direct head-on comparison [87]. The uncontrolled nature of the single-arm studies in which the data were pooled from precluded any comparison, thus further high-quality evidence is crucial. A multi-centre randomised controlled trial of early versus late EVD is in the planning stage. 

### 3.4. Surgical Technique

#### 3.4.1. Landmarks

External ventricular drains are most frequently inserted using the right Kocher’s point as an anatomical landmark. The Kocher’s point is located 11 cm posterior to the nasion in adults; equivalent to 1 cm anterior to the coronal suture, and 2 to 3 cm lateral to the midline, which is equivalent to the mid-pupillary line (Figure 4) [88]. Other access points have been described for ventricular access in trauma cases, including more anterior points, e.g., Ghajar point located 10 cm posterior to the nasion and 3 to 4 cm lateral to the midline [89,90]. After trephination at Kocher’s point, the ventricular catheter should follow a trajectory aiming at the ipsilateral medial epicanthus in the coronal plane and ipsilateral external auditory meatus at the sagittal plane [90,91]. There are, however, controversies regarding the optimal trajectory; a simpler trajectory, whereby the catheter is directed perpendicular to the skull surface, has been proposed [92]. This issue is further discussed in Section 3.4.4 and Section 3.4.5. The catheter tip should be 5 to 6 cm from the inner table of the calvarium, inserted in the frontal horn of the lateral ventricles or anterior to the Foramen of Monro [90].

#### 3.4.2. Tunnelling Recommendations

Upon ventricular access and optimal catheter positioning, the distal end of the catheter should be subcutaneously tunnelled before exposure. A longer distance from the burr hole has been associated with lower catheter-related infection due to a reduction of CSF leakage and ascending bacterial colonisation of the catheter from the catheter exit site [93]. However, existing evidence remains inconclusive as to whether longer tunnelling reduces infection. A prospective study conducted in Pakistan reported that long-tunnelled EVD delayed infections but with no significant difference regarding infection risk [94]. A recent prospective multi-centre cohort study in the UK and Ireland found no association between tunnelling distance and infection risk [95]. The heterogeneity of study population, protocols of EVD management, and definition of infection should be considered when comparing results [93,94,95]. 

#### 3.4.3. Bolt EVDs

Bolt EVDs are secured with a bolt, therefore obviating the need for tunnelling [96]. This has some theoretical advantages, such as lower risk of CSF leak, infection, and catheter migration [97]. Using the same technique and landmarks previously described, the EVD is inserted freehand, and then the bolt is screwed into the burr hole and secured. A recent prospective study in Denmark, including 32 tunnelled and 17 bolt EVDs, found that bolt EVDs reduce the incidence of overall complications compared to tunnelled EVDs (17.6% versus 59.4%; *p* = 0.007) [98]. However, when considering the risk of individual complications, bolt EVDs did not confer a statistically significant advantage in reducing infection risk. The incidence rate per EVD treatment day for infections were 1.3% higher in the tunnelled group [98]. This concurs with previous studies [99]. The low number of included EVDs may explain the lack of statistical significance for this complication type. Nonetheless, there are large retrospective studies demonstrating their accuracy, safety, and cost effectiveness [96,97,98,99]. This is a potentially promising technology to reduce the risk of complications, but ultimately, randomised trials will be needed to reach robust conclusions. 

#### 3.4.4. Accuracy of EVD Placement

Freehand EVD insertion carries the risk of multiple placement attempts and inaccuracy, especially in patients with small ventricles [100,101,102,103]. Published literature on the freehand insertion technique for EVDs includes patients with mixed indications for EVD insertion, but patients with sTBI often make up less than 50% of the cohort [101,102]. This subpopulation may increase the difficulty of such an insertion, as sTBI patients often have collapsed or smaller ventricles, or a midline shift with displaced ventricles, both of which are associated with higher rates of misplacement [101,103,104]. Successful insertion, which can be determined by the presence of free-flowing CSF through the catheter, does not always correlate with accurate EVD placement [100]. In single-centre studies determining the accuracy of EVD placement in the ICU and operating rooms, EVD catheter tips have been observed in eloquent parenchyma, such as basal ganglia and internal capsule, extraventricular space, non-ventricular CSF spaces, and the contralateral lateral ventricle [100,103]. Such misplacements result in more frequent replacement or repositioning, especially if the CSF is not draining, with the risk of complications and additional brain injury increasing with each attempt [100,102]. In addition, catheter correction represents an opportunity cost in terms of lost time for therapeutic drainage, and the additional time and cost of repeated CT scans forgone for other patients [100]. In 2008, a grading system for ventricular catheters’ tip location was proposed by neurosurgeons of the Barrow Neurological Institute in Phoenix, Arizona (USA), where they analysed CT scans of 346 patients that underwent EVD placement (Figure 5) [105]. In this case series, 77% of all EVDs placed were optimal, 10% were suboptimal in non-eloquent tissue, and 13% were suboptimal in eloquent tissue. Whether a tunnelled or a bolt-connected EVD is used may also affect the accuracy of EVD placement, as a retrospective Denmark study demonstrated a 20% increase of patients with optimal catheter placement in the latter group [97]. 

#### 3.4.5. Devices to Improve Accuracy

Several devices and techniques have been applied to improve placement accuracy. The Ghajar device (Neurodynamics, Inc., New York, NY, USA) and the Thomale Guide (Miethke GbmH & Co Kg, Potsdam, Germany) guides the catheter along a path perpendicular to the cranial surface into the ipsilateral frontal horn of the lateral ventricle when placed over the burr hole (Figure 6) [89,100]. Due to their reliance on normal ventricular anatomy, such devices may not be superior to the freehand technique in cannulating displaced ventricles [103,106]. A 2008 survey of 934 neurosurgeons showed that only 3% of the respondents regularly used guides, whilst 92% employ the freehand technique [32]. Increased procedural time, unfamiliarity with the technology, or the belief of the adequacy and safety of the current practice are potential reasons [32]. Other techniques to aid visualisation and improve placement efficiency involve image guidance using ultrasonography or computed tomography [107]. 

In a retrospective cohort study comparing the freehand technique, stereotactic neuronavigation, and ultrasound guidance, only 55% of the freehand-passed EVDs were accurately placed, in contrast to 88% and 89% for the latter techniques, respectively [107]. However, it is noteworthy that TBI patients only made up 1% of the entire cohort, thus these more time-consuming entities may not be applicable to these patients considering the emergency situations they present in [107,109]. Despite demonstrable evidence of better accuracy and lower morbidity, similar usage data to the Ghajar guide were reported for image guidance, with 1% of the respondents indicating regular use [32]. There may have been less resistance to embrace these new technologies in the 11 years that have elapsed, yet existing data are limited to single institutions [101,107,110]. Recently, a 3D virtual reality-simulated study performed at the University of Bern in Switzerland projected different access points in the anterior area of 50 CTs of patients with normal ventricle anatomy [111]. After simulation of several trajectories, the most successful ones were achieved at 10 cm from the nasion and 3 to 5 cm laterally from the midline, directing the catheter to the nasion or the contralateral medial canthus [111].

The recent evidence-based consensus statement from the Neurocritical Care Society recommends the use of image guidance in cases of aberrant ventricular anatomy if available but recognises the low-quality evidence and the lack of meaningful comparison groups [112]. As such, a prospective randomised trial of a portable, easy-to-use, CT-based smartphone device versus freehand-passed ventriculostomy is underway [113]. 

### 3.5. Complications

#### 3.5.1. Ventriculostomy-Associated Infections (VAIs)

Insertion of external ventricular drains carries the risk of complications, with infections being the most common; the pooled incidence rate was estimated to be 11.4 per 1000 catheter-days in one meta-analysis [114] although such studies are complicated by a lack of clinically reliable diagnostic criteria and the low sensitivity of the gold standard investigation; the identification of organisms in CSF microscopy or culture. Of note, the proportion for catheters placed for TBI was, on average, 28.0%. A retrospective analysis of 584 severely head-injured patients revealed several risk factors associated with VAIs: Predisposing diagnosis (intraventricular haemorrhage, operative depressed skull fracture); neurosurgical operations, such as craniotomy; and systemic infections (sepsis, pneumonia) showed significant correlation [115]. The finding that the duration of catheterisation increases the risk of VAIs concurred with existing literature, but the relationship was non-linear [95,115,116]. VAIs are associated with increased morbidity and mortality, longer hospitalisations, and higher healthcare costs, albeit the difficulty in ascertaining the relative contribution of underlying neurological events and the nosocomial infections to patients’ clinical outcomes [117]. 

Several preventative measures, such as prophylactic antibiotic administration, infection control protocols, and antibiotic-impregnated or silver-bearing catheters, have been utilised in an attempt to decrease VAI risk. In particular, uncertainty and variation of use over antibiotic therapies persist [95]. Some adopt the more aggressive prophylactic approach, which include prolonged systemic antibiotics. A recent meta-analysis demonstrated the efficacy of combined extended (>24 h) intravenous antibiotic administration and antibiotic-coated catheters [118]. When used in isolation, they outperformed perioperative administration of prophylactic antibiotics [118]. However, there are limitations when comparing studies in the existing literature. Notably, the variability in VAI definitions, which include positive CSF cultures or ventriculo-meningitis, makes it difficult to distinguish from contamination by skin flora [114,116]. The method and frequency of CSF sampling, as well as the clinical symptoms that may be confounded by the primary and secondary neurological insults, also adds to the challenge [114,116]. On the other hand, there are recommendations against extended prophylaxis [119]. First, increased risk of *Clostridium difficile* diarrhoea and the development of antibiotic-resistant organisms have been documented [119]. Second, there are concerns over a microbiological spectrum shift from gram-positive to more noxious gram-negative micro-organisms due to colony selection pressure [95,117,120]. Yet, this shift was not observed in a single-centre study in the UK, where they compared VAIs between two cohorts in 2006 and 2012, though it should be noted that susceptibility patterns are hospital- and region-dependent [120]. 

The efficacy of antibiotic- and silver-bearing catheters continues to be the subject of clinical debate. Despite heterogeneity and insufficient data, three meta-analyses to date, which included data from four randomised controlled trials, have pointed to its protective effect over VAIs [118,121,122]. However, a prospective multi-centre study in the UK showed that a third of EVD procedures in the UK and Ireland still use plain catheters [95]. They also demonstrated no significant difference in infection risk between plain, antibiotic-impregnated, and silver-bearing catheters. The contrasting results may be explained by the differences in baseline VAI risk between the available literature and the included institutions in the study [118]. Multi-centre RCTs similar to the BASICS (The British Antibiotic and Silver Impregnated Catheters for ventriculoperitoneal Shunts) trial would be beneficial in assessing the comparative effectiveness of the different catheter types.

#### 3.5.2. Ventriculostomy-Associated Haemorrhage

EVD placement and removal carry the risk of haemorrhage. Historically, reported rates of haemorrhage were low. However, this may be an under-estimation due to factors contributing to missed diagnoses, such as isodense haemorrhagic lesions in the fogging phase, confounding pre-existent neurological dysfunction; and small punctate haematomas that are overlooked especially with radio-opaque catheters [123,124]. The development of haemorrhage in TBI patients may be attributable to direct trauma to the vasculature during catheter placement, catheter adherence to the choroid plexus and enclosing parenchyma, or less strict guidelines on coagulation parameters in the case of EVD removal [125,126]. For the vast majority of patients, these haemorrhages have minimal clinical significance, which is defined as neurological deterioration on clinical examination temporally attributable to ventriculostomy [125,127]. However, an operation for evacuation may be required in some. Importantly, studies have observed the small volume of ventriculostomy-related haemorrhages in TBI patients [127,128]. Whilst the small size makes a mass effect unlikely, it could damage the parenchyma, cause neurocognitive changes, and form a seizure focus [125]. Compared with non-TBI population, patients with TBI were found to have significantly smaller volumes of intraparenchymal haemorrhage than patients with cerebrovascular diseases, 2 cm^3^ versus 11 cm^3^, in a retrospective review of 160 patients (TBI; *n* = 36) [127]. The significance was not replicated in a retrospective review of 276 patients with 360 EVD procedures (TBI; *n* = 16) [128]. It is worth noting that the relative risk of ventriculostomy-related haemorrhage cannot be definitively concluded. In both studies, several factors are at play: The small sample size of the respective conditions; the unreported influence of intracranial hypertension, which confers a predisposition to spontaneous haemorrhages; and the unmentioned composition of the trauma population. The relative proportion of coagulopathic TBI patients with a history of alcohol abuse may, for instance, influence the complication rates [124]. 

Neurosurgical dictum has sought to minimise haemorrhagic events through the use of plasma transfusions to normalise coagulation parameters, until the Internationalised Normalised Ratio (INR) is less than 1.2 to 1.4 [126,129,130], although this does not completely determine the functional coagulation profile of the patient. Yet, correcting coagulation parameters has been shown to delay neurosurgical interventions, with a risk of clinical deterioration [126,131]. A retrospective review in the US has demonstrated that the incidence of haemorrhage in TBI patients is not significantly different in those with an INR between 1.4 and 1.6 when compared to those with an INR < 1.2 and an INR between 1.2 and 1.4 [126]. The limitations of INR, the problems related to plasma transfusions (cost and risk of blood-borne illnesses, cardiac overload, and acute lung injury from transfusions), and the unpredictability of coagulopathy post-neurosurgery in trauma patients may contribute to varying practice patterns [131]. 

### 3.6. Paediatric Studies

#### 3.6.1. ICP Monitoring

ICP monitoring is recommended as a level III evidence in the current BTF guidelines for paediatric sTBI [132]. The choice of monitoring modality is not specified. Some studies report the preferred use of EVD due to the added benefit of CSF drainage [133,134]. On the other hand, a 2006 prospective national study in the UK and Ireland reported that EVDs were rarely used [135]. Potential reasons for this have been suggested, including the technical feasibility limited by the structure of the infant skull, as well as the appropriateness for individual patients based on clinical assessment [136,137]. ICP monitoring is done infrequently in some settings. A national review of paediatric patients in the US with a 7.7% (318/4141) rate of monitor use delineated the association between ICP monitoring and longer ICU and hospital length of stay (LOS), as well as the duration of ventilation [137]. Whist this may suggest the selective use of ICP monitoring in children with more severe injuries, the association of ICP monitoring and a longer ICU LOS were also confirmed in another study [135]. Patient- and hospital-level variability have also been documented, whereby older children treated in adult-only trauma centres are more likely to receive ICP monitoring [138].

#### 3.6.2. CSF Diversion

Ventricular CSF drainage is often employed in paediatric sTBI. The 2019 consensus guidelines for paediatric patients made a level III recommendation for CSF drainage as a means to control ICP [132]. The companion algorithm, which does not exist for adult TBI patients, recommends continuous CSF drainage as the initial therapeutic intervention [139]. However, the three studies that provided the evidence for the level III recommendation substantially differ with regards to the indications for drainage and drainage details. This makes comparing or aggregating results challenging. Shapiro and Marmarou used CSF drainage as a first-line therapy, initiated when ICP >20 mmHg with progressive increments or lasting for >10 min, or if plateau waves or spontaneous elevations of ICP >30 mmHg without associated noxious stimuli were observed [140]. Jagannathan et al. recruited children who either received medical management alone or medical management with surgery, ventriculostomy, or DC [141]. Ventriculostomy was performed in view of elevated ICP despite medical management, or when candidates were not suitable for DC. Andrade et al. investigated patients with posttraumatic diffuse brain swelling who underwent continuous CSF drainage [142]. The drainage strategy of continuous or intermittent was not reported in the former two case series, whilst continuous was used for the lattermost. It could influence results, since continuous drainage has been demonstrated to remove larger volumes of CSF, and achieve markedly lower ICP, MAP, and CPP in 19 paediatric patients [28]. The significantly lower concentration of CSF biomarkers in continuous drainage may reduce the duration and impact of the post-traumatic inflammatory process, thus reducing oedema [142]. 

There are studies on ICP control post-drainage in cohorts of mixed ages, but the assumption that the effect should be similar in adults and children should not be impetuously made due to age-related differences [143]. Paediatric patients with sTBI are more likely to develop diffuse cerebral swelling as a response to injury, with a smaller intracranial CSF volume to be displaced [144,145,146]. CSF drainage may thus be insufficient to allow adequate cerebral perfusion. Higher quality evidence will hopefully be available following completion of ADAPT (Approaches and Decisions in Acute Paediatric TBI Trial) [147].

### 3.7. EVDs in Resource-Limited Settings

In resource-limited settings, commercially manufactured EVD sets are not always accessible and affordable. Improvisations using readily available apparatuses to monitor ICP and temporarily drain CSF have been reported in the literature with success and minimal complications [148,149,150]. At hospitals in Nigeria and India, feeding tubes have been reported as substitutes to EVD catheters whilst a sterile urine collection bag completes the closed drainage system [148,150]. In an observational study in a Tunisian hospital, a 14-gauge central catheter assembled on a metal guide acted as a catheter, with the suture wings aiding fixation post-tunnelling [149]. Whilst post-operative complications mentioned in Section 3.5 can still occur, these makeshift EVDs may be useful [149]. Nevertheless, this highlights the issue that affordable equipment should be a priority target for universal health coverage; for example, in the field of hydrocephalus, affordable shunts have been manufactured and have been found to be effective [151].

### 3.8. Economic Considerations

In the prevailing climate of rising healthcare demand and tighter budgets, the cost-effectiveness of EVDs is an important consideration. The overall economic burden of TBI is substantial, with costs due to disability and lost productivity estimated to outweigh those for acute medical care and rehabilitation [152]. Nonetheless, existing literature focuses on the acute direct cost borne by EVDs in various conditions, mostly SAH or hydrocephalus, which include the cost of EVD-related complications (notably infections and haemorrhage), CT imaging, and supplies following placement failure or EVD obstruction [117,153,154]. Relatively less attention is placed on the long-term medical costs or indirect cost, such as loss of productivity, which is significant given the high incidence of TBI in young adults.

None investigated the cost-effectiveness of EVDs specifically for TBI. Whilst the additional cost of EVD-related complications may be similar, the health gains (benefit) from EVD should be considered separately. First, is EVD more cost-effective than other forms of monitoring, most commonly IPM? Second, is CSF drainage cost-effective? This is uncertain: On one hand, CSF drainage may not be an efficient means to reduce ICP, as TBI patients often have poor intracranial compliance [104]. On the other hand, TBI patients with CSF drainage have been reported to have reduced therapy intensity level, which saves the cost of other advanced ICP-lowering interventions, such as decompressive craniectomy and its subsequent cranioplasty [76,81]. In a cost-utility analysis of different treatment strategies in sTBI from a US societal perspective, aggressive care (compliance rate of >50% with BTF guidelines, including the use of invasive intracranial pressure monitoring and decompressive craniectomy) is the most cost-effective across all populations [155]. Compared to comfort care (one day of aggressive care without invasive monitoring or DC in the ICU followed by care on the medical-surgical floor) or routine care (BTF compliance rate <50%), the aggressive approach has the highest initial cost but the lowest lifetime costs due to substantial improvements in the patients [155]. The type of invasive intracranial pressure monitor used in the decision-analytical model was not mentioned, but the BTF guidelines used at the date of publication advised EVD as the first choice due to the accuracy, reliability, and cost [33].

## 4. Conclusions

EVDs allow the control and monitoring of ICP to guide treatment and the therapeutic drainage of cerebrospinal fluid to control ICP. This can improve perfusion and mitigate the risk of exacerbating secondary cerebral injury, as well as reduce the probability of brain herniation in TBI patients. Despite the routine use in tiered ICP-based management protocols, evidence for its use is scant and of low quality. Most of the available evidence on EVDs instead comes from studies investigating populations with subarachnoid haemorrhage or hydrocephalus. Much clinical debate exists over the facets of EVD usage, but the lack of standardisation, particularly in the definition of tiers and thresholds, poses a difficulty for comparative effectiveness research or systematic reviews to inform national guidelines. These should be clarified in ongoing and future prospective studies. 

## Figures and Tables

**Figure 1 jcm-08-01422-f001:**
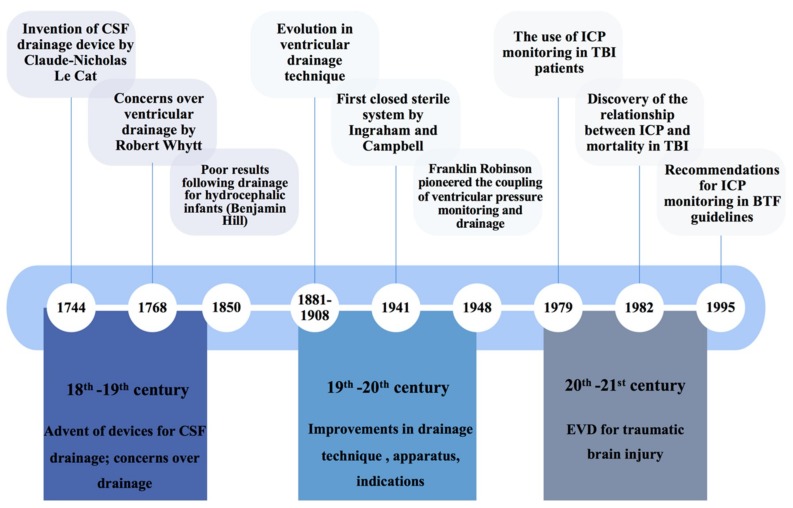
Timeline of selected events in the history of external ventricular drainage. Source: authors.

**Figure 2 jcm-08-01422-f002:**
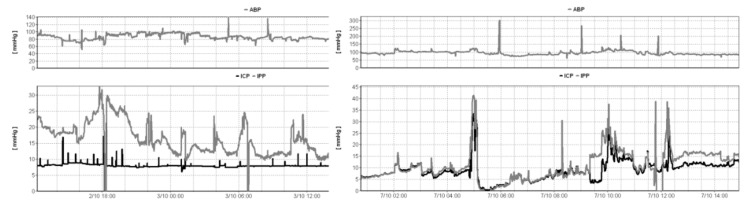
A recording featuring arterial blood pressure (ABP) and intraparenchymal pressure (IPP—bottom pane, grey line) together with EVD pressure (ICP—bottom panel, black line) using an external transducer in patient after poor grade aneurysmal subarachnoid haemorrhage. The left panel demonstrates the results with the drain opened, whereas the right panel demonstrates results with the drain closed. With an open EVD, the two pressure readings failed to correlate. EVD pressure is held constant at a value representing the calibrated level of the drain above the heart. With a closed EVD (right panel), the two measured pressure values correlate over time.

**Figure 3 jcm-08-01422-f003:**
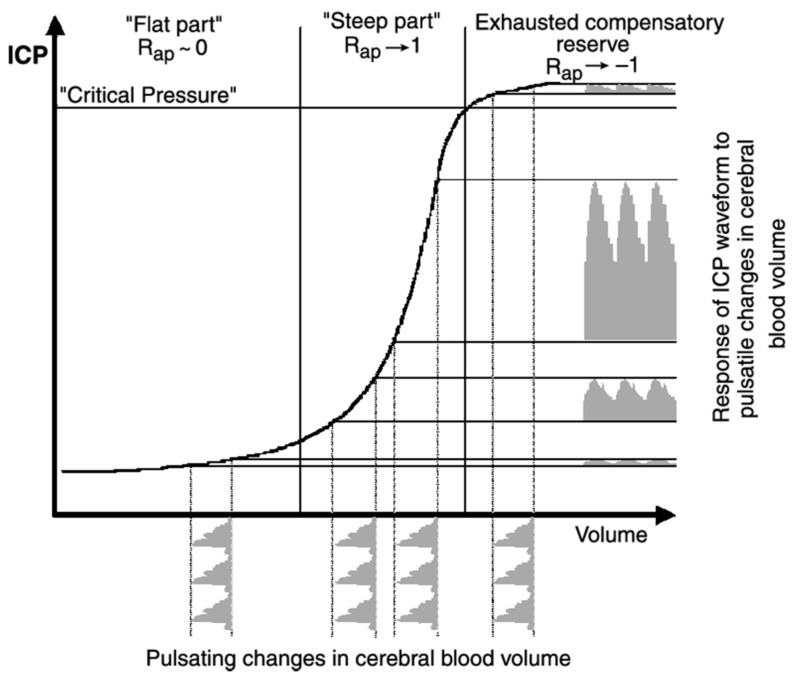
Sigmoidal curve delineating the intracranial pressure–volume relationship. Three distinct phases are shown: (i) Compensatory phase (“flat part”) represented by high intracranial compliance and low ICP due to physiological buffer systems; (ii) “steep part”: Decompensation phase marked by exponential increases in ICP with small increases in intracranial volume due to low compensatory reserve, indicated by the R_ap_ index; (iii) phase with exhausted compensatory reserve and deranged cerebrovascular reactivity. TBI patients with intracranial hypertension are often in the second or third phase. Figure taken from Steiner et al. [73].

**Figure 4 jcm-08-01422-f004:**
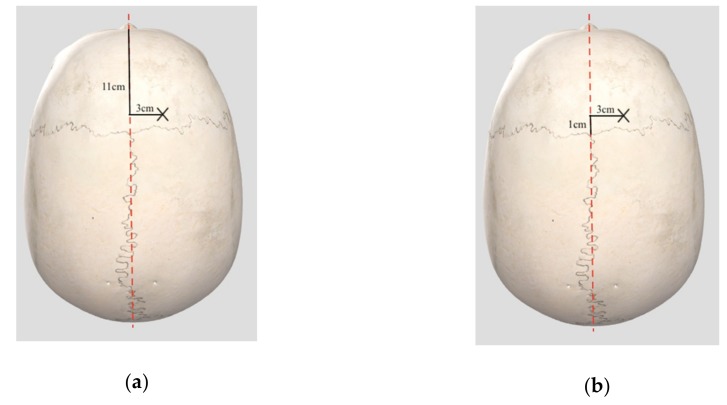
Kocher’s point (marked by “X”) indicated by surface anatomy. (**a**) Located 11 cm posterior to nasion and 3 cm lateral to the midline (red dotted line); (**b**) 1 cm anterior to the coronal suture and 3 cm lateral to the midline (red dotted line). Skull image obtained from 3D4Medical Complete Anatomy 2018 Version 3.3.0 ^®^.

**Figure 5 jcm-08-01422-f005:**
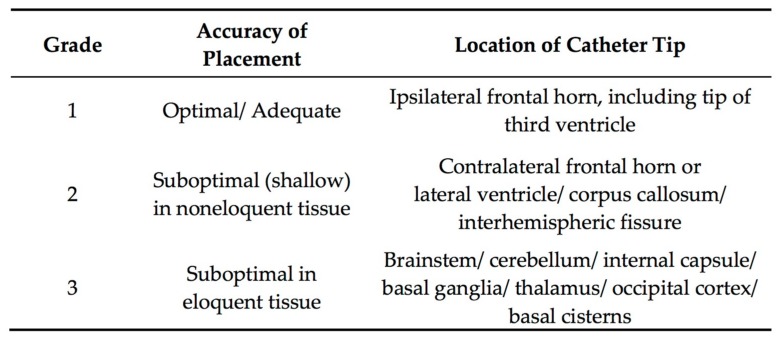
Grading system for catheter tip location. Figure taken from Kakarla et al. [105].

**Figure 6 jcm-08-01422-f006:**
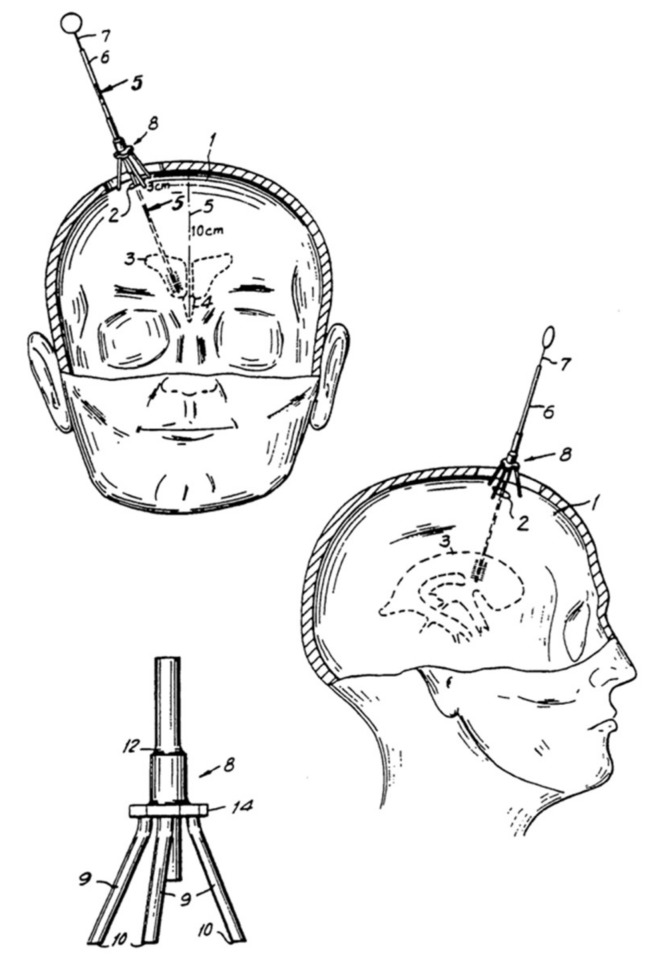
Recommendation of entry point with the Ghajar device (10 cm from nasion and 3 to 4 cm lateral to the midline). From: European Patent Office. Ghajar J: Apparatus for Guiding Catheter into Cerebral Ventricle. EP:0229105B1 (22/07/87) [108].

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
