# Peer review of "The Evolution of the Role of External Ventricular Drainage in Traumatic Brain Injury"

_jcm, 2019, doi:10.3390/jcm8091422_

Round 1

Reviewer 1 Report

This manuscript is a narrative review of the literature on the utilization of external ventricular drainage on traumatic brain injury, focusing on indications of therapeutic efficacy related to ICP lowering, as well as optimal strategies for drainage and timing of insertion. The authors present an historical overview, indications for eVD trains in TBI, surgical techniques, complications, paediatric studies, and economic considerations. The lack of standardization in the protocols for EVD usage creates difficulties in comparing efficacy amongst studies. They suggest that this should be clarified in ongoing and future prospective studies, but do not propose standards.  The major conclusion is that although EVDs are routinely used in tiered ICP-based management protocols, there is a paucity of evidence to support such use and the existing evidence is of low quality. Overall, this is a comprehensive and well-written review. Comments are relatively minor.

Comments:

1. The authors acknowledge that this is a narrative review and not a systematic review or meta-analysis, and therefore a PRISMA flow chart is not necessary. Nonetheless, additional information on the review criteria would be helpful. The authors indicate that “Selection of articles for inclusion in our Review was based on a combination of research quality, high number of citations, and current and potential impact on clinical practice.”  However, they also indicate that “Evidence to support the use of CSF drainage in TBI patients is limited and of low quality, and is largely derived from uncontrolled observational studies (Table 1).” This begs the question of how research quality used as a selection criteria for articles.  This, and the other criteria for article selection, should be expanded upon.

2. The search criteria may have been too restrictive. A PubMed search using the indicated search strategy (“traumatic brain injury” AND  “external ventricular drains” AND “raised ICP” yields a single hit. Thus, it would be helpful to indicate how many articles were identified from the databases searched, how many were read, and how many were considered in the review and how many were discarded because of low research quality or other criteria.

3. In Table 1, the significance of the bold font for some results should be explained.

4. Although a section on paediatric TBI is included, the results presented in Table 1 do not indicate whether the studies are adult TBI, paediatric TBI, or a combination.

5. Minor: The contribution of several authors appears to be reviewing and editing the manuscript. These contributions could be considered in an acknowledgement rather than as authorship.

Reviewer 2 Report

Extensive and thorough manuscript on external ventricular drainage in traumatic brain injury. Some aspects may be somewhat long. However, there is not much to add, all important aspects are considered. Maybe one question could also be considered: Is there any connection between continuous CSF-drainage over several days and development of posttraumatic hydrocephalus? There is some discussion on stimulating CSF-production by CSF-drainage via EVD.
